# Juglone Suppresses LPS-induced Inflammatory Responses and NLRP3 Activation in Macrophages

**DOI:** 10.3390/molecules25133104

**Published:** 2020-07-07

**Authors:** Nam-Hun Kim, Hong-Ki Kim, Ji-Hak Lee, Seung-Il Jo, Hye-Min Won, Gyeong-Seok Lee, Hyoun-Su Lee, Kung-Woo Nam, Wan-Jong Kim, Man-Deuk Han

**Affiliations:** Department of Life Science and Biotechnology, College of Natural Sciences, Soonchunhyang University, Asan, Chuncheongnam-do 31538, Korea; sem5603@gmail.com (N.-H.K.); pgswkt@naver.com (H.-K.K.); marong932@naver.com (J.-H.L.); 309cho@naver.com (S.-I.J.); whm1223@naver.com (H.-M.W.); ssm4914@sch.ac.kr (G.-S.L.); lhs21c77@hanmail.net (H.-S.L.); kwnam1@sch.ac.kr (K.-W.N.); wjkim56@sch.ac.kr (W.-J.K.)

**Keywords:** juglone, NLRP3 inflammasome, caspase-1, IL-1β, IL-18

## Abstract

The NLRP3 (NACHT, LRR and PYD domains-containing protein 3) inflammasome has been implicated in a variety of diseases, including atherosclerosis, neurodegenerative diseases, and infectious diseases. Thus, inhibitors of NLRP3 inflammasome have emerged as promising approaches to treat inflammation-related diseases. The aim of this study was to explore the effects of juglone (5-hydroxyl-1,4-naphthoquinone) on NLRP3 inflammasome activation. The inhibitory effects of juglone on nitric oxide (NO) production were assessed in lipopolysaccharide (LPS)-stimulated J774.1 cells by Griess assay, while its effects on reactive oxygen species (ROS) and NLRP3 ATPase activity were assessed. The expression levels of NLRP3, caspase-1, and pro-inflammatory cytokines (IL-1β, IL-18) and cytotoxicity of juglone in J774.1 cells were also determined. Juglone was non-toxic in J774.1 cells when used at 10 μM (*p* < 0.01). Juglone treatment inhibited the production of ROS and NO. The levels of NLRP3 and cleaved caspase-1, as well as the secretion of IL-1β and IL-18, were decreased by treatment with juglone in a concentration-dependent manner. Juglone also inhibited the ATPase activities of NLRP3 in LPS/ATP-stimulated J774.1 macrophages. Our results suggested that juglone could inhibit inflammatory cytokine production and NLRP3 inflammasome activation in macrophages, and should be considered as a therapeutic strategy for inflammation-related diseases.

## 1. Introduction

Innate immune cells, including macrophages and dendritic cells, play crucial roles in the initiation of inflammatory immune responses upon activation of inflammasome complexes that induce the maturation of inflammatory cytokines [1]. Pattern-recognition receptors (PRRs) recognize pathogen-associated molecular patterns (PAMPs) and damage-associated molecular patterns (DAMPs), initiating signaling cascades that lead to inflammation and release of inflammatory cytokines [2,3]. Toll-like receptors (TLRs) and NOD-like receptors (NLRs) called PRR are membrane-bound and cytoplasmic receptors, respectively, which recognize PAMPs and DAMPs [4]. The immune responses initiated by the interaction between TLRs and PAMPs are mediated through MyD88-dependent and TRIF-dependent pathways [2,3,5]. When TLRs recognize and bind lipopolysaccharides (LPS) or other extracellular PAMPs, the transcription factor nuclear factor (NF)-κB is activated and induces the expression of several pro-inflammatory cytokines, including interleukin (IL)-1β and IL-18 [6]. On the other hand, NLRs, PRRs found in the cytoplasm, are activated by DAMPs, including adenosine triphosphate (ATP), monosodium urate crystals, β-amyloid, and nigericin [7].

P2X7, a membrane-bound PPR that recognizes DAMP, is a membrane-bound receptor that binds extracellular ATP, promoting the release of K^+^ [8,9], and subsequently, inducing the formation of inflammasomes after binding of NLRP3 (NACHT, LRR and PYD domains-containing protein 3) to ASC (apoptosis-associated speck-like protein containing a C-terminal caspase recruitment domain) and pro-caspase-1. The formation of NLRP3 inflammasomes activates pro-caspase-1, which induces pyroptosis by cleaving pro-IL-1β and pro-IL-18 into their active forms, which initiate the innate immune response [7]. In addition, intracellular danger signals, such as mitochondrial damage and reactive oxygen species (ROS) production, can promote assembly and activation of NLRP3 inflammasomes [10,11,12]. Inflammatory factors and cytokines can also prime NLRP3 inflammasomes, and ROS inhibitors have been reported to suppress NLRP3 priming [13]. Therefore, the treatment of macrophages with potent antioxidative agents has emerged as a promising strategy to block NLRP3 inflammasome priming.

Juglone (5-hydroxy-1,4-naphthoquinone) is a natural phenolic compound isolated from the roots, stems, and leaves of walnut trees [14]. Juglone is produced during allelopathy and inhibits the germination or growth of surrounding plants by inhibiting specific enzymes necessary for metabolic function, mitochondrial respiration, and photosynthesis [15]. Recent studies have reported that juglone has a variety of pharmacological effects, including anti-inflammatory, antioxidative, anticancer, and antimicrobial effects, by inhibiting ROS-producing enzymes and preventing oxidative stress [14,16,17,18,19,20,21]. However, the effect of juglone on NLRP3 inflammasome formation and activation remains elusive. The aim of this study was to investigate the potential of juglone to inhibit the NLRP3 inflammasome-mediated inflammatory effects of LPS and ATP.

## 2. Results

### 2.1. Effects of Juglone on Cell Viability

To determine the juglone concentration range that is non-toxic to J774.1 cells, cells were treated with increasing concentrations of juglone (3.1–50 μM), followed by 3-(4,5-dimethylthiazol-2-yl)-2,5-diphenyltetrazolium bromide (MTT) assay. When juglone was used at 10 μM or less, no significant cytotoxic effects were observed (*p* > 0.05; Figure 1). Therefore, juglone was used at 2.5, 5, or 10 μM in subsequent experiments.

### 2.2. Effects of Juglone on Reactive Oxygen Species (ROS) Production

ROS have a strong oxidative capacity and are essential mediators of NLRP3 (NACHT, LRR and PYD domains-containing protein 3) inflammasome activation. To determine the ability of juglone to inhibit lipopolysaccharide (LPS)-induced ROS production in J774.1 cells, we used the cell-permeable ROS-sensitive dye DCFH-DA, which is non-fluorescent in a reduced state and emits fluorescence upon oxidation by ROS. We found that treatment with LPS enhanced ROS production in J774.1 cells. However, when cells were treated with 10 μM juglone, ROS production was significantly impaired (Figure 2a). The fluorescence intensity in LPS-treated cells was approximately 120%, while the fluorescence intensity in juglone-treated (10 μM) cells after LPS stimulation was only 84.5%. Thus, juglone may exert anti-inflammatory effects by suppressing ROS generation by macrophages during inflammation.

### 2.3. Effects of Juglone on Nitric Oxide (NO) Production

Inflammatory mediators, such as reactive nitrogen species and ROS, are essential mediators of inflammatory responses, especially in the initial stages of inflammation. Therefore, we assessed the effects of juglone in LPS-induced NO production. We found that even though LPS treatment resulted in a significant increase in NO production in J774.1 cells, juglone treatment suppressed LPS-induced NO production in a dose-dependent manner (*p* < 0.05; Figure 2b).

### 2.4. Effects of Juglone on the Secretion of the Pro-Inflammatory Cytokines IL-1β and IL-18

Inflammatory cytokines are important mediators of immune responses and orchestrate the initial stages of inflammation. The pro-inflammatory cytokines IL-1β and IL-18 are known to mediate the first steps of an inflammatory immune response [22]. Therefore, targeting these upstream mediators of inflammation has emerged as a promising approach to treat inflammation-related diseases. Thus, we sought to investigate whether juglone could affect the levels of mRNA, protein, or extracellular secretion of IL-18 and IL-1β. We found that treatment with juglone suppressed the induction of IL-18 and IL-1β at the mRNA and protein level in response to LPS and ATP in J774.1 cells (Figure 3 and Appendix A). The secretory inhibition of pro-inflammatory cytokines (IL-1β and IL-18) of juglone on murine macrophage cells was studied and the results are shown Figure 2c,d. Treatment of macrophage cells with juglone caused a concentration-dependent reduction in their IL-1β (5 µM, * *p* < 0.05) and IL-18 (10 µM, ** *p* < 0.01). Juglone-treated cell was reduced 25.9% of IL-1β secretion (5 µM, * *p* < 0.05) and 22.6% of IL-18 secretion (10 µM, ** *p* < 0.01) compared to the LPS plus STP-primed control group. These results suggested that juglone could inhibit the initial steps of the inflammatory immune response by suppressing the expression and secretion of the pro-inflammatory cytokines IL-1β and IL-18 in macrophages.

### 2.5. Effects of Juglone on NLRP3 and Caspase-1 Expression

We then assessed the effects of juglone treatment on the mRNA and protein levels of NLRP3 and caspase-1. We found that even though NLRP3 was induced in J774.1 cells following stimulation with LPS and ATP, treatment with juglone suppressed the LPS/ATP-mediated NLRP3 induction at the mRNA level in a dose-dependent manner. The mRNA levels of caspase-1 were not significantly affected by juglone treatment (Figure 4a). Importantly, juglone treatment suppressed the LPS/ATP-mediated NLRP3 ATPase upregulation at the protein level in J774.1 macrophages (Figure 4b and Appendix A). Juglone also reduced the levels of cleaved caspase-1 in a concentration-dependent manner (Figure 4b). These results suggested that juglone could inhibit the interaction of NLRP3 and apoptosis-associated speck-like protein containing a C-terminal caspase recruitment domain (ASC) and subsequent NLRP3 inflammasome formation by downregulating the expression of NLRP3. Moreover, the decreased NLRP3 expression might impair ASC speck formation, leading to reduced cleavage and subsequent activation of the downstream signaling mediator, caspase-1.

### 2.6. Effects of Juglone on the ATPase Activity of NLRP3

The pyrin ATP-binding domain of NLRP3 exhibits ATPase activity, which is essential for NLRP3 inflammasome oligomerization. Therefore, we investigated the effects of juglone treatment on the ATPase activity of NLRP3 in J774.1 cells. Treatment with LPS and ATP in J774.1 macrophages increased the ATPase activity of NLRP3 (Figure 5). However, treatment with 10 μM of juglone significantly reduced the ATPase activity of NLRP3 (*p* < 0.05). 

## 3. Discussion

The NLRP3 (NACHT, LRR and PYD domains-containing protein 3) inflammasome can be activated by a broad range of stimuli that belong either to pathogen-associated molecular patterns (PAMPs) released during viral, bacterial, fungal, or protozoa infection [23] or to danger-associated molecular patterns (DAMPs) of endogenous or exogenous origin, like extracellular ATP and reactive oxygen species (ROS) [24]. The present study demonstrates that juglone inhibits IL-1b and IL-18 secretion in activated macrophages by suppressing various pro-inflammatory signaling molecules and pathways.

Juglone has been reported to have potent antioxidative effects and prevent various oxidative stress-related diseases by inhibiting ROS-producing enzymes [25,26]. Additionally, juglone has been suggested to have anticancer effects, which are least partly mediated by inhibition of NF-κB signaling and NF-κB-mediated expression of inflammatory cytokines [27,28]. However, the effects of juglone NLRP3 inflammasome formation in macrophages are understudied.

IL-1β, IL-18, and NO have been highlighted by numerous studies as essential mediators of inflammation and have been implicated in autoimmune diseases and other inflammatory conditions [29]. LPS, a major component of Gram-negative bacteria, can induce the expression of inflammatory cytokines and NO. Herein, we reported that juglone treatment inhibited the production of NO in LPS-stimulated J774.1 cells in a concentration-dependent manner. ROS are also involved in inflammatory immune responses and have been shown to induce formation and activation of the NLRP3 inflammasome in response to various exogenous stimuli [30]. In addition, excessive intracellular ROS levels can lead to apoptosis or necrosis [31]. To investigate the effects of juglone on ROS production by macrophages, a DCFH-DA assay was performed in J774.1 cells treated with LPS. Our results suggested that LPS stimulation induced ROS production and that juglone could suppress LPS-induced ROS generation in a dose-dependent manner. Therefore, by reducing ROS production in macrophages, juglone could potentially inhibit activation of the NLRP3 inflammasome.

ATP-induced P2X7R activation promotes the rapid production of large amounts of ROS, which, in turn, stimulates activation of the NLRP3 inflammasome [32]. It has been previously reported that inhibition of the ATPase activity of NLRP3 could decrease the self-oligomerization of NLRP3, as well as its interaction with ASC, which is critical for inflammasome activation [31].

In this study, we found that juglone treatment suppresses not only the expression of NLRP3 but also the ATPase activity of NLRP3, and these were followed by the inhibition of active caspase-1 (Figure 6). These results suggest that juglone could potentially impair the formation of NLRP3 inflammasome, and, thus, the inhibition of pro-caspase-1 activation in macrophages. We also found that treatment with juglone could suppress the LPS/ATP-mediated induction of IL-1β and IL-18 in J774.1 cells, both at the mRNA and protein levels. These correspond to the previous research results that juglone inhibits pro-inflammatory cytokines (TNF-α, IL-1 β, and IL-6) and adhesion molecules (VCAM-1 and ICAM-1) expression through the inhibition of IκB-phosphorylation-mediated NF-κB activation [17,33].

Furthermore, our extended study results showed that the secretion levels of IL-1β and IL-18 in macrophages were inhibited by juglone treatment in a dose dependent manner. Of note, pro-IL-1β and pro-IL-18 are cleaved to their active forms by caspase-1 upon activation by NLRP3 inflammasome. The expression of IL-18 showed relatively small change rather than that of IL-1β. It may be due to distinct regulation for Il-1β and IL-18. It is reported that there are distinct licensing requirements for processing of IL-1β and IL-18 activation by NLRP3 inflammasome in mice [34]. Considering that juglone is natural product, it is valuable that juglone shows anti-inflammatory effects in 10 µM of concentration. Because juglone might be categorized as a potential anticancer compound based on the criteria established by the National Cancer Institute [35] that any compound with IC_50_ value of ≤ 4 µg/mL has the potential to be an anticancer compound. In view of these present findings and the existing reports of its use in traditional folk medicine as an anti-inflammatory agent, juglone deserves further studies to justify its potential as an anti-inflammatory agent, using a spectrum of preclinical models. Hence, our results indicate that juglone treatment could reduce the production of mature inflammatory cytokines IL-1β and IL-18 by suppressing their transcription and translation levels as well as caspase-1-mediated cleavage to their active forms. Therefore, our findings suggest that juglone could be used as a potential therapeutic compound for treating inflammatory diseases in the future. 

## 4. Materials and Methods

### 4.1. Chemical Compounds

Juglone (5-hydroxy-1,4-naphthoquinone; purity > 97%) was purchased from Merck (Darmstadt, Germany). Adenosine 5′-triphosphate (ATP) and lipopolysaccharide (LPS) were purchased from Sigma-Aldrich, Inc. (St. Louis, MO, USA).

### 4.2. Cell Cultures

J774.1 cells were purchased from the Korean Collection for Type Cultures and maintained in DMEM (Gibco BRL, Gaithersburg, MD, USA) supplemented with 10% fetal bovine serum (Gibco BRL), 100 µg/mL streptomycin, and 100 U/mL penicillin (Gibco BRL). Cells were incubated at 37 °C in a humidified 5% CO_2_ incubator. J774.1 cells were subjected to the following treatments: (i) control; (ii) LPS (Escherichia coli O111:b4) + ATP; (iii) 2.5 μM juglone + LPS + ATP; (iv) 5 μM juglone + LPS + ATP; (v) 10 μM juglone + LPS + ATP. For these treatments, cells were incubated in the presence of juglone for 2 h, followed by treatment with LPS (1 μg/mL) for 6 h and ATP (5 mM) for 1 h. After treatment, cells were harvested for further analyses.

### 4.3. Cell Viability

The cytotoxic effects of juglone were assessed by 3-(4,5-dimethylthiazol-2-yl)-2,5-diphenyltetrazolium bromide (MTT) assay. Briefly, J774.1 cells were seeded in 24-well plates (2 × 10^5^ cells/well) in 500 μL of cell culture medium and incubated in a 37 °C and 5% CO_2_ incubator for 12 h to allow for cell adhesion. After incubation, cells were treated with increasing concentrations of juglone (3.1–50 μM) for 24 h. Subsequently, the cell culture medium was replaced with phenol red-free medium, and 50 μL of 5 mg/mL MTT solution was added to each well. Cells were incubated with MTT for an additional 4 h in a 37 °C and 5% CO_2_ incubator. The supernatant was removed, and 200 μL dimethyl sulfoxide (DMSO) was added to each well to dissolve the MTT crystals. Optical absorbance was measured at 450 nm using a microplate reader (Marshall Scientific, Hants, UK). Cell viability was calculated, according to the following Formula (1):Cell viability (%) = (Abs 450 nm of untreated cells − Abs 450 nm of treated cells/Abs 450 nm of untreated cells) × 100(1)

### 4.4. Measurement of Nitric Oxide (NO)

To evaluate the anti-inflammatory effects of juglone in macrophages, NO production was measured in J774.1 cells. The amount of NO produced in response to LPS stimulation was measured using the Griess test, which determines the amount of nitrite (NO_2_^−^) generated. J774.1 cells were seeded in 24-well plates (2 × 10^5^ cells/well) and incubated at 37 °C for 12 h to allow for cell adhesion. After incubation, the cell culture medium was replaced with phenol red-free medium. Cells were treated with different concentrations of juglone for 2 h, followed by stimulation with LPS (1 μg/mL) for 24 h at 37 °C and 5% CO_2_. After incubation, 200 μL of the cell culture solution and 50 μL of Griess reagent [1% (*w*/*v*) sulfanilamide and 0.1% N-1-naphthylethylene diamine in 2.5% (*v*/*v*) phosphoric acid] were mixed in a 96-well plate. After a 10 min incubation at room temperature, the absorbance was measured at 540 nm. The standard calibration curve was obtained by measuring the optical absorbance of serial dilutions of sodium nitrate (NaNO_2_). The amount of NO produced was determined using the standard calibration curve.

### 4.5. Measurement of the Antioxidative Capacity of Juglone

The antioxidative capacity of juglone was measured using the hydrophilic peroxyl radical scavenging capacity (PSC) assay [36]. To measure ROS in J774.1 cells, 2,7-Dichlorodihydrofluorescein diacetate (DCFH-DA, Sigma, St. Louis, MO, USA) was used. The cells were seeded into 24-well plates (2 × 10^5^ cells/well) and incubated for 12 h. Cells were treated with 2.5, 5, and 10 μM juglone. After 2 h of incubation, cells were exposed to 1 μg/mL LPS for 24 h in a 37 °C and 5% CO_2_ incubator. After incubation, the cells were washed twice with cold phosphate-buffered saline (PBS) and incubated with 10 μM of DCFH-DA for 30 min at 37 °C. The fluorescence intensity was measured on a fluorescence microplate reader (Marshall Scientific) using an excitation wavelength of 485 nm and an emission wavelength of 535 nm.

### 4.6. Measurement of Cytokine Release

The amounts of IL-1β and IL-18 produced by J774.1 cells were measured using enzyme-linked immunosorbent assay (ELISA). Briefly, cells were seeded into a 24-well plate (1 × 10^6^ cells/well) and incubated for 12 h in a 37 °C and 5% CO_2_ incubator. Subsequently, cells were treated with 2.5, 5, and 10 μM of juglone for 2 h, followed by LPS treatment (1 μg/mL) for 6 h and ATP treatment (5 mM) for 1 h. The production of pro-inflammatory cytokines was assessed using commercial ELISA kits (mouse IL-1β: R&D Systems, Minneapolis, MN, USA; mouse IL-18: Cloud-Clone Co., Katy, TX, USA) according to the manufacturer’s instructions. The absorbance was measured at 540 nm. All experiments were performed in triplicate.

### 4.7. RNA Isolation and cDNA Synthesis

Total cellular RNA from J774.1 cells was isolated using the RNeasy Midi Kit (Qiagen GmbH, Hilden, Germany) according to the manufacturer’s instructions. Samples were treated with RNase-free DNase I (Takara, Dalian, China) at 37 °C for 30 min to avoid any DNA contamination. The quality and quantity of the RNA samples were measured using a bioanalyzer (Agilent 2100; Agilent Technologies, Santa Clara, CA, USA). RNAs with RNA integrity values ≥ 8.0 were used. Complementary DNA was synthesized from 1 μg total RNA using the iScript cDNA synthesis kit (Bio-Rad Laboratories, Hercules, CA, USA). cDNA synthesis was performed at 46 °C for 20 min, followed by enzyme inactivation at 95 °C for 1 min.

### 4.8. Real-Time Reverse Transcriptase Quantitative Polymerase Chain Reaction (RT-qPCR)

Total RNA was extracted using TRIzol reagent (Life Technologies, Carlsbad, CA, USA) according to the manufacturer’s instructions. RT-qPCR was performed on a CFX96 Real-time PCR (Bio-Rad Laboratories Inc., Hercules, CA, USA). RT-qPCR reactions were prepared using 1 μL cDNA, 10 μL SYBR Green master mix (Bio-Rad Laboratories, Inc.), and 2 μL of primer mix in a total volume of 20 μL. Thermocycling conditions were as follows: 3 min at 95 °C, 35 cycles of denaturation (15 s at 95 °C), and combined annealing/extension (30 s at 60 °C). GAPDH was used as a housekeeping gene. Primer sequences and their respective PCR fragment lengths were as follows: NLRP3 (NACHT, LRR and PYD domains-containing protein 3), forward 5′-GTGGAGATCCTAGGTTTCTCTG-3′ and reverse 5′-CAGGATCTCATTCTCTTGGATC-3′; Caspase-1, forward 5′-GAGCTGATGTTGACCTCAGAG-3′ and reverse 5′-CTGTCAGAAGTCTTGTGCTCTG-3′; IL-1β, forward 5′-GTGGAGATCCTAGGTTTCTCTG-3′ and reverse 5′-CAGGATCTCATTCTCTTGGATC-3′; IL-18, forward 5′-AGGACACTTTCTTGCTTGCC-3′, and reverse 5′-CACAAACCCTCCCCACCTAA-3′; GAPDH forward 5′-GTGGGGCGCCCCAGGCACCA-3′ and reverse 5′-CTCCTTAATGTCACGCACGATTTC-3′. After amplification, a melting curve (0.01 °C/s) was used to confirm product purity. Relative mRNA content was normalized to GAPDH content.

### 4.9. Western Blot Analysis

Cells were seeded into 6-well plates (2 × 10^6^ cells/well) and incubated in a 37 °C and 5% CO_2_ incubator for 12 h. Cells were then treated with different concentrations of juglone for 2 h, followed by treatment with LPS (1 μg/mL) for 6 h and ATP (5 mM) for 1 h. Cells were then washed with cold PBS and lysed with RIPA buffer (Thermo Scientific, Waltham, MA, USA). Protein concentration was quantified using a BCA protein assay kit (Sigma). Proteins (40 μg) were separated by SDS-PAGE (10% and 15% gels) and transferred onto polyvinylidene difluoride membranes using a Mini Tran-Blot Electrophoretic Transfer Cell (Bio-Rad Laboratories). Membranes were blocked with 5% skimmed milk at room temperature for 2 h, and washed three times with 1× TBST buffer (20 mM Tris-HCl, 150 mM NaCl, and 5% Tween 20, pH 7.6). Membranes were incubated with the primary antibodies (1:1000 dilution) at 4 °C for at least 10 h. After three washes with 1 × TBST buffer, membranes were incubated with mouse or rabbit IgG-horseradish peroxidase conjugated secondary antibodies at room temperature for 2 h. Westsave ECL detection reagent and a chemiluminescent Western blot imaging instrument (Labgear Australia, Milton, Australia) were used to visualize the signal intensities. NLRP3 (NACHT, LRR and PYD domains-containing protein 3; #13158), and caspase-1 antibody (#24232) were purchased from Cell Signaling (Danvers, MA, USA). IL-1β (sc-7884) and β-Actin (sc-130656) antibody were bought from Santa Cruz Biotechnology (Dallas, TX, USA). IL-18 antibody (A16737) was purchased from ABclonal (Woburn, MA, USA). Mouse anti-rabbit secondary antibody (sc 2357)/or anti-mouse IgG HRP secondary antibody (sc516102) were bought from bought from Santa Cruz Biotechnology (Dallas, TX, USA). All antibodies were used at a dilution of 1:1000.

### 4.10. ATPase Activity Measurement

J774.1 cells were seeded into 6-well plates (2 × 10^6^ cells/well) and incubated at 37 °C for 12 h. Cells were then treated with juglone for 2 h, followed by treatment with LPS (1 μg/mL) for 6 h and ATP (5 mM) for 1 h. After washing twice with 0.9% NaCl, lysis buffer (40 mM Tris, 80 mM NaCl, 8 mM MgAc2, and 1 mM EDTA, pH 7.5) was used to lyse the cells. Cells were centrifuged at 15,000 rpm for 15 min at 4 °C. After pushing the cell lysate through a 25-gauge needle, the aqueous layer was collected and analyzed using the ATPase activity assay kit (Sigma), according to the manufacturer’s instructions.

### 4.11. Statistical Analysis

All the data were analyzed using SPSS Statistics 25 software (International Business Machines, Armonk, NY, USA). Data are expressed as mean ± standard deviation. Statistical significance was determined using one-way ANOVA, and *P* < 0.05 was considered statistically significant; * *p* < 0.05 and ** *p* < 0.01 

## 5. Conclusions

Increased ROS, NO, and NLRP3 ATPase activity by LPS and/or ATP treatment causes activation of the NLRP3 inflammasome in macrophages. Juglone treatment reduced the production of ROS and NO, and inhibited the ATPase activity of NLRP3 in J774.1 macrophages. Moreover, juglone treatment suppressed NLRP3 expression and inhibited NLRP3 inflammasome activation, thereby inhibiting the cleavage of caspase-1. Secretion of the pro-inflammatory cytokines IL-1β and IL-18 was also decreased by juglone in a concentration-dependent manner. Thus, juglone should be considered as a therapeutic strategy for inflammation-related diseases.

## Figures and Tables

**Figure 1 molecules-25-03104-f001:**
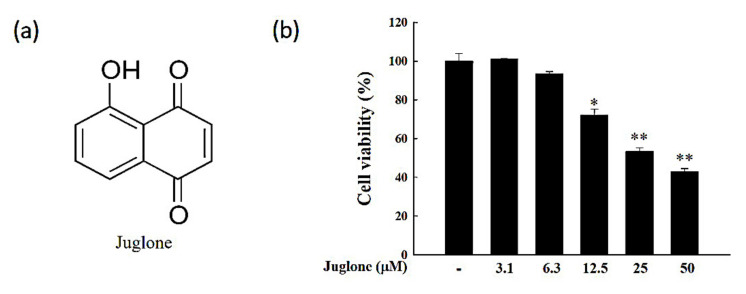
Assessment of the cytotoxic effects of juglone in J774.1 cells using the 3-(4,5-dimethylthiazol-2-yl)-2,5-diphenyltetrazolium bromide (MTT) assay. (**a**) Chemical structure of juglone. (**b**) J774.1 cells were treated with increasing concentrations of juglone (3.1–50 μM) for 24 h, followed by quantification of cell viability using the MTT assay. Data are presented as mean ± standard deviation (SD) of three independent experiments. * *p* < 0.05; ** *p* < 0.01; vs. control.

**Figure 2 molecules-25-03104-f002:**
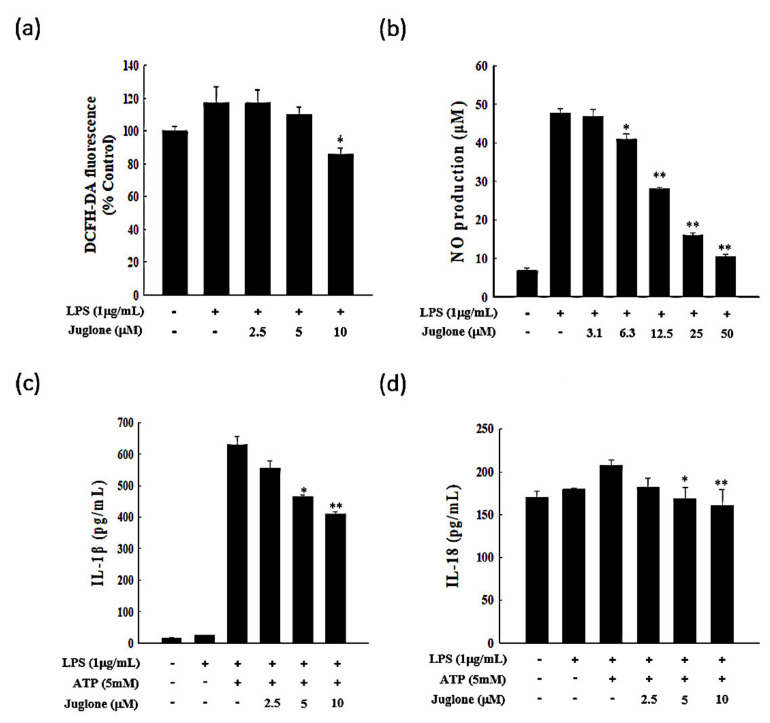
Effects of juglone on the production of inflammatory mediators. (**a**) reactive oxygen species (ROS) production in J774.1 cells treated with lipopolysaccharide (LPS). Cells were treated with different concentrations of juglone (2.5, 5, and 10 μM) for 2 h, followed by treatment with 1 µg/mL LPS for an additional 24 h. (**b**) NO production in J774.1 cells treated with LPS. Cells were treated with increasing concentrations of juglone (3.1–50 μM or 0 μM control) for 2 h, followed by treatment with 1 μg/mL LPS for an additional 24 h. The levels of NO in the cell culture media were measured using Griess reagent. (**c**,**d**) Effects of juglone on the LPS/ATP-induced secretion of interleukin-1β (IL-1β) and IL-18 in J774.1 cells. J774.1 macrophages were treated with different concentrations of juglone for 2 h, followed by treatment with 1 μg/mL LPS for 6 h and treatment with 5 mM ATP for 1 h. The levels of IL-1β and IL-18 secreted in the culture medium were analyzed by enzyme-linked immunosorbent assay (ELISA). Data are presented as mean ± SD of three independent experiments. * *p* < 0.05; ** *p* < 0.01 vs. LPS or LPS + ATP cells, respectively.

**Figure 3 molecules-25-03104-f003:**
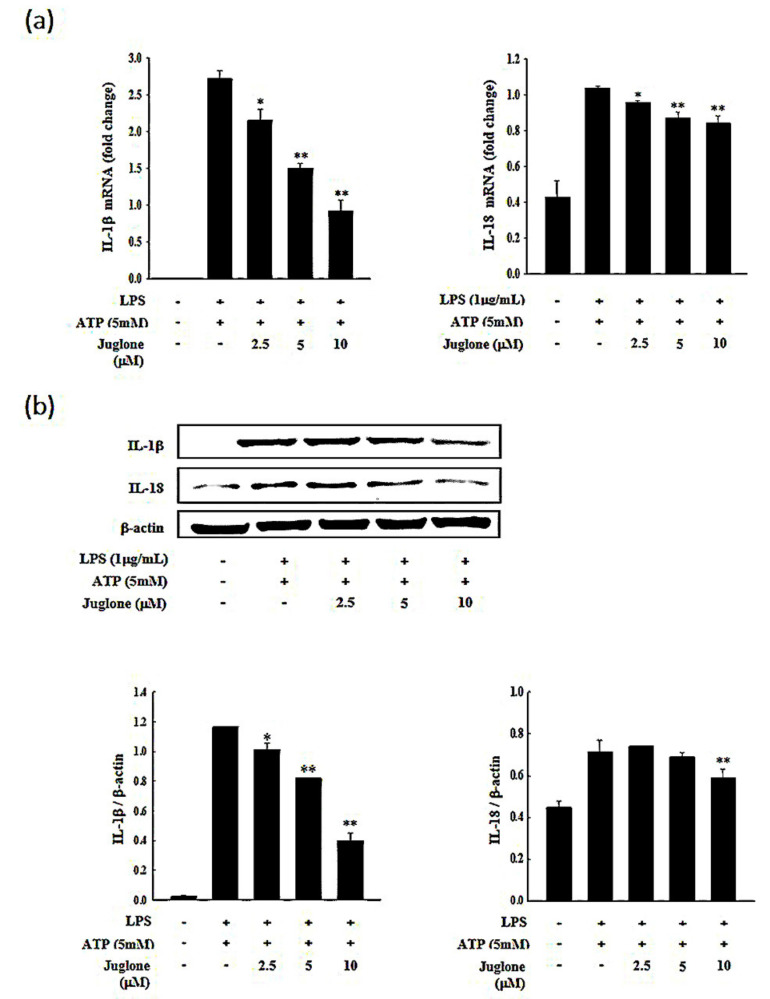
Effects of juglone on IL-1β and IL-18 expression in LPS-treated J774.1 cells. J774.1 macrophages were treated with different concentrations of juglone (2.5, 5, and 10 μM) for 2 h, followed by treatment with 1 μg/mL LPS for 6 h and 5 mM ATP for 1 h. (**a**) The mRNA levels of IL-1β and IL-18 in J774.1 cells were determined with RT-qPCR. Glyceraldehyde 3-phosphate dehydrogenase gene (GAPDH) was used as a reference gene. (**b**) The protein levels of IL-1β and IL-18 in J774.1 cells were assessed by Western blotting and are presented as relative to β-actin intensity. Data are presented as mean ± SD. * *p* < 0.05; ** *p* < 0.01 vs. LPS + ATP treated cells.

**Figure 4 molecules-25-03104-f004:**
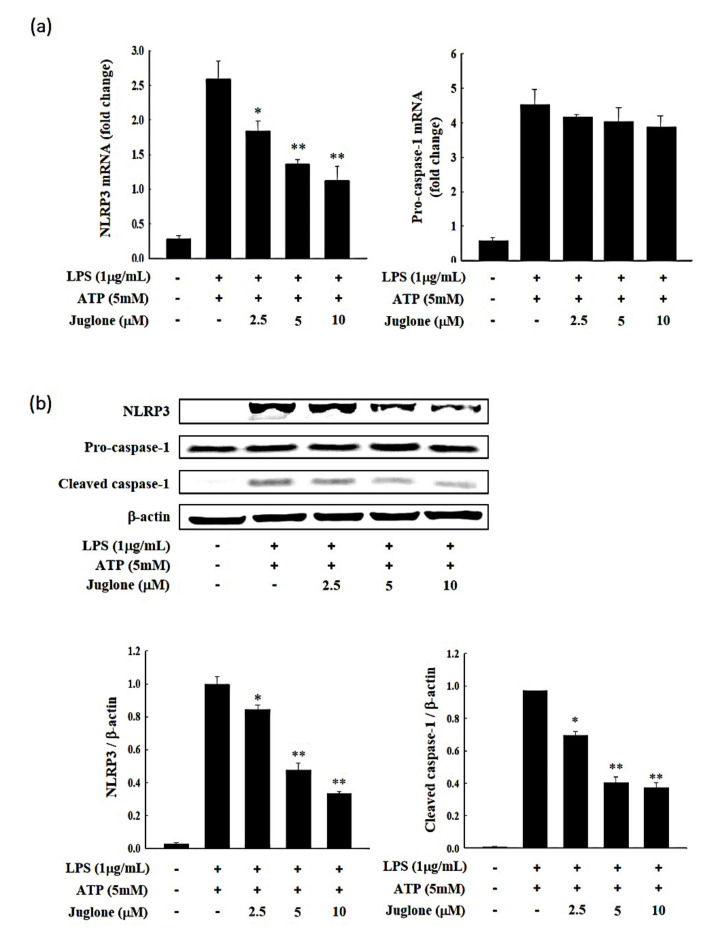
J774.1 cells were treated with various concentrations of juglone for 2 h and then treated with 1 μg/mL LPS for 24 h. (**a**) The mRNA levels of NACHT, LRR and PYD domains-containing protein 3 (NLRP3) and caspase-1 in J774.1 cells were determined by RT-qPCR. GAPDH was used as a reference gene. (**b**) The relative protein levels of NLRP3, procaspase-1, and cleaved caspase-1 were assessed by Western blotting and are presented as relative to β-actin intensity. Data are presented as the mean ± SD.* *p* < 0.05; ** *p* < 0.01 vs. LPS + ATP treated cells.

**Figure 5 molecules-25-03104-f005:**
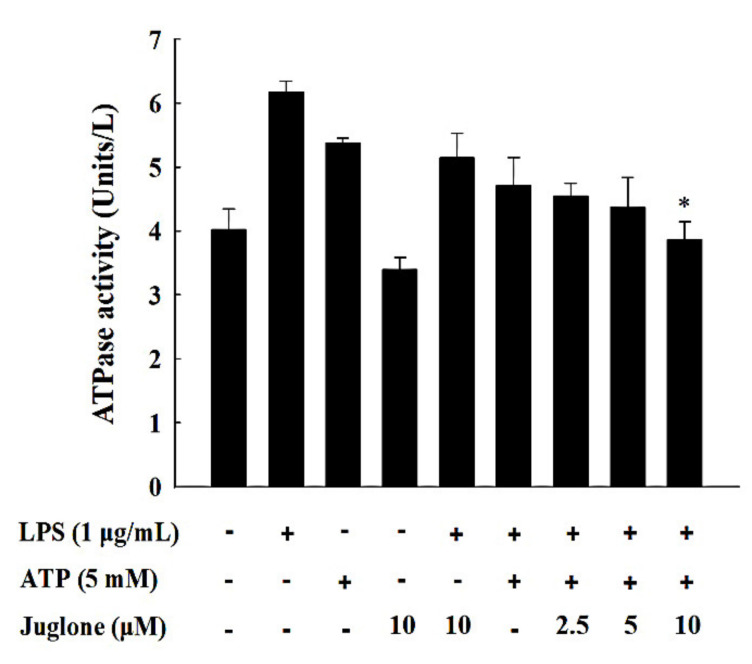
Effects of juglone on the ATPase activity of NLRP3 in LPS/ATP-treated J774.1 cells. J774.1 macrophages were treated with different concentrations of juglone for 2 h, followed by treatment with 1 μg/mL LPS for 6 h and 5 mM ATP for an additional hour. Analysis of the ATPase activity of NLRP3 was performed using a reaction mixture containing 40 mM Tris, 80 mM NaCl, 8 mM MgAc2, 1 mM EDTA, and 4 mM ATP, pH 7.5. Data are presented as mean ± SD of three independent experiments. * *p* < 0.05 vs. LPS/ATP treated cells.

**Figure 6 molecules-25-03104-f006:**
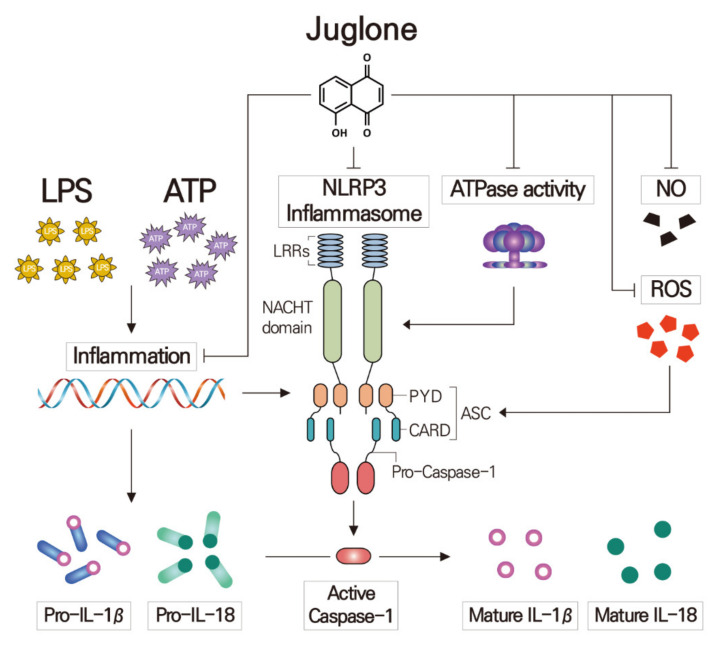
Proposed mechanism underlying the anti-inflammatory effects of juglone in macrophages. Pre-treatment of macrophages with juglone suppressed ATP-induced IL-1β, IL-18 and NLRP3 secretion in LPS-primed J774.1 mouse macrophages. Juglone also caused downregulation of ATPase activity and ROS and NO production. Juglone also reduced the mRNA expression and activation of IL-1β, IL-18, NLRP3 and caspase-1. The results show that juglone inhibits IL-1β and IL-18 secretion and NLRP3 formation in activated macrophages.

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
