# Peer review of "Juglone Suppresses LPS-induced Inflammatory Responses and NLRP3 Activation in Macrophages"

_molecules, 2020, doi:10.3390/molecules25133104_

Round 1

Reviewer 1 Report

22/0/2020

Juglone suppresses LPS-induced inflammatory responses and NLRP3 activation in macrophages.

There is sufficiently novelty in the study to provide a stronger argument of why this study was conducted. The authors report a simple and precise study into the Juglone suppresses LPS-induced inflammatory responses and NLRP3 activation in macrophages. The manuscript is a well-written manuscript. The work is well conducted with adequate introduction, methodology and results; however, I feel the authors have not dealt with the discussion section in depth. So, I recommend authors must add more in discussion sections to substantiate their findings.

Author Response

Responses to comments from Review #1

  1. General comments

Juglone suppresses LPS-induced inflammatory responses and NLRP3 activation in macrophages. There is sufficiently novelty in the study to provide a stronger argument of why this study was conducted. The authors report a simple and precise study into the Juglone suppresses LPS-induced inflammatory responses and NLRP3 activation in macrophages. The manuscript is a well-written manuscript. The work is well conducted with adequate introduction, methodology and results; however, I feel the authors have not dealt with the discussion section in depth. So, I recommend authors must add more in discussion sections to substantiate their findings.

Response: Thank you for the review. We appreciate your insightful suggestions. We re-checked the Discussion section and revised it to add some content for further review. Please find the revised sentences in the Discussion section as following:

NLRP3 inflammasome can be activated by a broad range of stimuli that belong either to pathogen-associated molecular patterns (PAMPs) released during viral, bacterial, fungal, or protozoa infection [23] or to danger-associated molecular patterns (DAMPs) of endogenous or exogenous origin, like extracellular ATP and reactive oxygen species (ROS) [24]. The present study demonstrates that juglone inhibits IL-1β and IL-18 secretion in activated macrophages by suppressing various pro-inflammatory signaling molecules and pathways.

ATP-induced P2X7R activation promotes the rapid production of large amounts of ROS, which, in turn, stimulates activation of the NLRP3 inflammasome [32]. It has been previously reported that inhibition of the ATPase activity of NLRP3 could decrease the self-oligomerization of NLRP3, as well as its interaction with ASC, which is critical for inflammasome activation [33].

In this study, we found that juglone treatment suppress not only the expression of NLRP3 but also the ATPase activity of NLRP3, and these were followed by the inhibition of active caspase-1. These results suggest that juglone could potentially impair the formation of NLRP3 inflammasome, and thus the inhibition of pro-caspase-1 activation in macrophages. We also found that treatment with juglone could suppress the LPS/ATP-mediated induction of IL-1β and IL-18 in J774.1 cells, both at the mRNA and protein levels. These correspond to the previous research results that juglone inhibit pro-inflammatory cytokines (TNF-α, IL-1β, and IL-6) and adhesion molecules (VCAM-1, ICAM-1) expression through the inhibition of IκB-phosphorylation-mediated NF-κB activation [17, 34].

Furthermore, our extended study results showed that the secretion levels of IL-1β and IL-18 in macrophages were inhibited by juglone treatment in a dose dependent manner. Of note, pro-IL-1β and pro-IL-18 are cleaved to their active forms by caspase-1 upon the activation by NLRP3 inflammasome. In our study, Il-18 showed relatively small change rather than that of Il-1 β. It may be due to distinct regulation for cytokines. It is reported that there are distinct licensing requirements for processing of IL-18 and IL-1β activation by NLRP3 inflammasomes in mice [36]. Hence, our results indicate that juglone treatment could reduce the production of mature inflammatory cytokines IL-1β and IL-18 by suppressing their transcription and translation levels as well as caspase-1-mediated cleavage to their active forms. Therefore, our findings suggest that juglone could be used as a potential therapeutic compound for treating inflammatory diseases in the future.

Man-Deuk Han, Ph.D.

(Corresponding author)

Reviewer 2 Report

The manuscript by Nam-Hun Kim et al. investigate the inhibitory effect of the natural polyphenol juglone on the formation of the inflammasome involving NALP3 in J774.1 cells exposed to LPS or ATP.

The manuscript appears motivated, well presented and written.

However, some inaccuracies in the introduction need to be corrected. On the other hand, some specifications are needed to make the introduction more readable.

Line 31:

Please, briefly explain what an inflammasome complexe is

Line 34:

Please, specify that TLRs and NLRs are PPRs

Line 40-42

Please, rewrite the sentence because there are inaccuracies:

“On the other hand, NLRs, PRRs found in the cytoplasm, are activated by DAMPS, including ATP, monosodium urate crystals, β-amyloid, and nigericin”.

Line 43

Please, specify that PSX7 is a membrane-bound PPR able to detect DAMPs

Line 44

Please, write the name of NALP3 in full.

NLRP3 is the gene encoding the protein “NACHT, LRR and PYD domains-containing protein 3” (NALP3), which is involved in the formation of the known “NLRP3 inflammasome”. Please correct.

Finally,

I suggest that authors add a figure showing the steps involved in the formation of the NLPR3 inflammasome. This figure would show the studied targets (ROS, caspase, NO,..) and the step of the inflammasome complexe influenced by tested polyphenol.

Author Response

Responses to comments from Review #2

  1. General comments

The manuscript by Nam-Hun Kim et al. investigate the inhibitory effect of the natural polyphenol juglone on the formation of the inflammasome involving NALP3 in J774.1 cells exposed to LPS or ATP. The manuscript appears motivated, well presented and written. However, some inaccuracies in the introduction need to be corrected. On the other hand, some specifications are needed to make the introduction more readable.

Response: We are grateful for your comments, and the positive evaluation of our work. We have revised and modified the text and Figures according to your critiques. Our responses to your detailed question is below

  1. Detailed comments

Line 31: Please, briefly explain what an inflammasome complexe is

Response: We agree that it is better to provide more details about the explanation of inflammasome complex in the Introduction section. So, we explained the term inflammasome as the following sentence. “Innate immune cells, including macrophages and dendritic cells, play crucial roles in the initiation of inflammatory immune responses upon activation of inflammasome complexes that induce the maturation of inflammatory cytokines”.

 Line 34: Please, specify that TLRs and NLRs are PPRs

Response: Thank you for noting this. We corrected it with the following sentence; “Toll-like receptors (TLRs) and NOD-like receptors (NLRs) called PRR are membrane-bound and cytoplasmic receptors, respectively, which recognize PAMPs and DAMPs.”

Line 40-42: Please, rewrite the sentence because there are inaccuracies:

“On the other hand, NLRs, PRRs found in the cytoplasm, are activated by DAMPS, including ATP, monosodium urate crystals, β-amyloid, and nigericin”.

Response: Thank you for your exact point. We revised the sentence in correct as your suggestion: “On the other hand, NLRs, PRRs found in the cytoplasm, are activated by DAMPS, including ATP, monosodium urate crystals, β-amyloid, and nigericin.”

Line 43: Please, specify that PSX7 is a membrane-bound PPR able to detect DAMPs

Response: We sincerely appreciate your insightful comments and suggestions for revising the paper. As you suggested, we have corrected and revised it in the revision as following: “P2X7, a membrane-bound PPR that recognizes DAMP, is a membrane-bound receptor that binds extracellular ATP, promoting the release of K+ and subsequently inducing the formation of inflammasomes after binding of NLRP3 (NACHT, LRR and PYD domains-containing protein 3) to ASC (apoptosis-associated speck-like protein containing a C-terminal caspase recruitment domain) and pro-caspase-1.”

Line 44: Please, write the name of NALP3 in full.

NLRP3 is the gene encoding the protein “NACHT, LRR and PYD domains-containing protein 3” (NALP3), which is involved in the formation of the known “NLRP3 inflammasome”. Please correct.

Response: We completely agree with comment. Full name of NLRP3 was certainly specified in first mentioned site in text as suggestion (NACHT, LRR and PYD domains-containing protein 3).

Finally,

I suggest that authors add a figure showing the steps involved in the formation of the NLPR3 inflammasome. This figure would show the studied targets (ROS, caspase, NO,..) and the step of the inflammasome complex influenced by tested polyphenol.

Response: We added the schematic diagram of our present study as Figure 6. Our manuscript was more improved by your comment. Proposed mechanism underlying the anti-inflammatory effects of juglone in macrophages is presented in a new Figure 6.

(Please find the Figure 6 in the attatched word file.)

Figure 6. Proposed mechanism underlying the anti-inflammatory effects of juglone in macrophages. Pre-treatment of macrophages with juglone suppressed ATP-induced IL-1β, IL-18 and NLRP3 secretion in LPS-primed J774.1 mouse macrophages. Juglone also caused down-regulation of ATPase activity, ROS and NO production. Juglone also reduced the mRNA expression and activation of IL-1β, IL-18, NLRP3 and caspase-1. The results show that juglone inhibits IL-1 β and IL-18 secretion and NLRP3 formation in activated macrophages.

Man-Deuk Han, Ph.D.

(Corresponding author)

Reviewer 3 Report

The experimental data reported here are definitely worth publishing, but a few minor aspects needs correction or improvement, as discussed below:

Lines 54-62: because the introduction should show the scientific background of the investigation, the previous research on the anti-inflammatory effects of juglone should be cited here, such as

Seetha A, Devaraj H, Sudhandiran G. Indomethacin and juglone inhibit inflammatory molecules to induce apoptosis in colon cancer cells. J Biochem Mol Toxicol. 2020;34(2):e22433. doi:10.1002/jbt.22433

Peng X, Nie Y, Wu J, Huang Q, Cheng Y. Juglone prevents metabolic endotoxemia-induced hepatitis and neuroinflammation via suppressing TLR4/NF-κB signaling pathway in high-fat diet rats. Biochem Biophys Res Commun. 2015;462(3):245-250. doi:10.1016/j.bbrc.2015.04.124 etc

Line 58: “juglone has a variety of pharmacological effects, including antioxidant,”. “Antioxidant” is not a pharmacological effect, but a (bio)chemical one, therefore in my view, this should be eliminated from the sentence.

Lines 75 - 84: The anti-inflammatory effect of juglone is interesting, but the authors should also discuss that the effect is only seen at the highest concentration evaluated (10 uM), which is relatively hard to be reached in the human tissues.

Lines 111-115: the authors should have determined not only the statistical significance of the differences, but also effect sizes, because it seems that there is an effect, but it is rather modest, particularly on IL-18, and rather seen at relatively high concentrations (> 6 uM).

Line 204: “(2 × 105 cells/well)” is most likely “(2 × 10^5 cells/well). Similarly, on line 271 and line 291, “(2 × 106 cells/well)” should be “(2 × 10^6 cells/well)”

Line 254 and the following: If I understand correctly, the authors have used reverse-transcriptase real time PCR. If my understanding is correct, then the name should be “real time RT-qPCR”.  

Lines 156-185: The discussions section should include at least one paragraph on the limitations of the study, for instance the fact that the effect was seen at rather relatively high concentrations, that the effect was rather small on IL-18 and the ATPase activity (as well as an attempt of an explanation on the difference in effects on the different targets assessed). The discussions should also better place the results in the context of the state of the art knowledge on juglone.

Author Response

Responses to comments from Review #3

  1. General comments

The experimental data reported here are definitely worth publishing, but a few minor aspects needs correction or improvement, as discussed below:

Response: We are grateful for your comments, and the positive evaluation of our work.

We rechecked our manuscript carefully and revised the parts you pointed out. Here is our response to your detailed question.

  1. Detailed comments

Lines 54-62: because the introduction should show the scientific background of the investigation, the previous research on the anti-inflammatory effects of juglone should be cited here, such as Seetha A, Devaraj H, Sudhandiran G. Indomethacin and juglone inhibit inflammatory molecules to induce apoptosis in colon cancer cells. J Biochem Mol Toxicol. 2020;34(2):e22433. doi:10.1002/jbt.22433

Peng X, Nie Y, Wu J, Huang Q, Cheng Y. Juglone prevents metabolic endotoxemia-induced hepatitis and neuroinflammation via suppressing TLR4/NF-κB signaling pathway in high-fat diet rats. Biochem Biophys Res Commun. 2015;462(3):245-250. doi:10.1016/j.bbrc.2015.04.124 etc

 Response: We appreciate your kind comment. We added two references for anti-inflammatory effects of juglone as your suggestion. In the introduction section, the manuscript was revised as follows: “Recent studies have reported that juglone has variety of pharmacological effects, including anti-inflammatory, antioxidative, anticancer, and antimicrobial effects, by inhibiting ROS-producing enzymes and preventing oxidative stress [16-21].

Line 58: “juglone has a variety of pharmacological effects, including antioxidant,”. “Antioxidant” is not a pharmacological effect, but a (bio)chemical one, therefore in my view, this should be eliminated from the sentence.

Response: We completely agree your point. Antioxidant in text should rewrite to ‘antioxidative’ to mean the biological effects. We changed the word in text to your comment.

Lines 75 - 84: The anti-inflammatory effect of juglone is interesting, but the authors should also discuss that the effect is only seen at the highest concentration evaluated (10 uM), which is relatively hard to be reached in the human tissues.

Response: We thank the reviewer for asking these important questions. We agree to your comment. The aim of our study was to identify novel anti-inflammatory agent from natural products and not to assess preclinical dose. So, we added the following sentence in the Discussion section. “Considering that juglone is natural product, it is valuable that juglone show anti-inflammatory effects in 10 µM of concentration. Because juglone might be categorized as a potential anticancer compound based on the criteria established by the National Cancer Institute (Pisha et al., 1995) that any compound with IC50 value of ≤ 4 µg/ml has a potential to be an anticancer compound. In view of these present findings, and the existing reports of its use in traditional folk medicine as an anti-inflammatory agent, juglone deserves further studies to justify its potential as an anti-inflammatory agent, using a spectrum of preclinical models.” Also, the natural products related with inhibition of NLRP3, IL-1β and IL-18 are focused on many studies. They were also studied in similar concentration and showed their anti-inflammatory effects (at the lower part). So we regard that juglone is valuable natural product.

Resveratrol (50~100 µM, Fu et al. 2013.; 1~100 µM, Huang et al. 2014.; 0~60 µM, Chang et al. 2015.); Mangiferin (0.1~10 µM, Song et al. 2015.; 12.5~50 µM, Pan et al. 2016.); Sulforaphane (1~5 µM, An et al. 2016.); Ginseng (5~20 µM, Yoon et al. 2015.); Aloe emodin (12.5~50 µM, Han et al. 2016.); Quercetin (25~100 µM, Wang et al. 2013.); Curcumin (0.1~10 µM, Li et al. 2015.; 10~50 µM, Gong et al. 2015.); Genipin (50~200 µM, Rajanbabu et al. 2015)

Lines 111-115: the authors should have determined not only the statistical significance of the differences, but also effect sizes, because it seems that there is an effect, but it is rather modest, particularly on IL-18, and rather seen at relatively high concentrations (> 6 uM).

Response: We agree with the reviewer that it is necessary to determine the anti-inflammatory effect degree and its statistical significance of juglone. We added the following sentence in the Results: “The secretory inhibition of proinflammatory cytokine (IL-1β and IL-18) of juglone on murine macrophage cells were studied and results showed Figure 2c, 2d. Treatment of macrophage cells with juglone caused a concentration-dependent reduction in their IL-1β (5 µM, * p <0.05) and IL-18 (10 µM, ** p <0.01). Juglone–treated cell was reduced 25.9% of IL-1β secretion (5 µM, *p<0.05) and 22.6% of Il-18 secretion (10 µM, ** p <0.01) compared to the LPS plus ATP-primed control group.”

Line 204: “(2 × 105 cells/well)” is most likely “(2 × 10^5 cells/well). Similarly, on line 271 and line 291, “(2 × 106 cells/well)” should be “(2 × 10^6 cells/well)”

Response: We are thankful that your detail comments. They were changed to superscripts in text.

Line 254 and the following: If I understand correctly, the authors have used reverse-transcriptase real time PCR. If my understanding is correct, then the name should be “real time RT-qPCR”.  

Response: Your recommendation is right. That is our missing. We changed to ‘real-time reverse transcriptase quantitative polymerase chain reaction (Real-time RT-qPCR)’ as your recommendation.

Lines 156-185: The discussions section should include at least one paragraph on the limitations of the study, for instance the fact that the effect was seen at rather relatively high concentrations, that the effect was rather small on IL-18 and the ATPase activity (as well as an attempt of an explanation on the difference in effects on the different targets assessed). The discussions should also better place the results in the context of the state of the art knowledge on juglone.

Response: We appreciate your insightful reviews. We rewritten the Discussion section as the revised manuscript. To explain "Reviewer Comment: the fact that the effect was rather small on IL-18 and the ATPase”, we added the sentences as following: “The expression of Il-18 showed relatively small change rather than that of Il-1β. It may be due to distinct regulation for Il-1β and Il-18. It is reported that there are distinct licensing requirements for processing of IL-18 and IL-1β by NLRP3 inflammasomes in mice [36].” The author’s responses on the limitations of the study showed author’s responses to Comments Lines 75 – 84 above.

Man-Deuk Han, Ph.D.

(Corresponding author)

Round 2

Reviewer 1 Report

Dear Editor,
Yes, authors have made substantial changes to the discussion section. I am satisfied with the revised version. It can be accepted now for the publication.